# Frustrated supercritical collapse in tunable charge arrays on graphene

Jiong Lu[1,2,3], Hsin-Zon Tsai[1,4,5], Alpin N. Tatan[3,6], Sebastian Wickenburg[1,5], Arash A. Omrani[1], Dillon Wong[1,5], Alexander Riss[1,10], Erik Piatti[1,7], Kenji Watanabe[8], Takashi Taniguchi[8], Alex Zettl[1,5,9], Vitor M. Pereira [3,6] & Michael F. Crommie [1,5,9]

The photon-like behavior of electrons in graphene causes unusual confinement properties that depend strongly on the geometry and strength of the surrounding potential. We report bottom-up synthesis of atomically-precise one-dimensional (1D) arrays of point charges on graphene that allow exploration of a new type of supercritical confinement of graphene carriers. The arrays were synthesized by arranging $F_4TCNQ$ molecules into a 1D lattice on back-gated graphene, allowing precise tuning of both the molecular charge and the array periodicity. While dilute arrays of ionized $F_4TCNQ$ molecules are seen to behave like isolated subcritical charges, dense arrays show emergent supercriticality. In contrast to compact supercritical clusters, these extended arrays display both supercritical and subcritical characteristics and belong to a new physical regime termed "frustrated supercritical collapse". Here carriers in the far-field are attracted by a supercritical charge distribution, but their fall to the center is frustrated by subcritical potentials in the near-field, similar to trapping of light by a dense cluster of stars in general relativity.

[1] Department of Physics, University of California at Berkeley, Berkeley, CA 94720, USA. [2] Department of Chemistry, National University of Singapore, 3 Science Drive 3, Singapore 117543, Singapore. [3] Centre for Advanced 2D Materials, National University of Singapore, 6 Science Drive 2, Singapore 117546, Singapore. [4] International Collaborative Laboratory of 2D Materials for Optoelectronic Science & Technology of Ministry of Education, Engineering Technology Research Center for 2D Material Information Function Devices and Systems of Guangdong Province, Shenzhen University, Shenzhen 518060, China. [5] Materials Sciences Division, Lawrence Berkeley National Laboratory, Berkeley, CA 94720, USA. [6] Department of Physics, National University of Singapore, 2 Science Drive 3, Singapore 117542, Singapore. [7] Department of Applied Science and Technology, Politecnico di Torino, Torino 10129 TO, Italy. [8] National Institute for Materials, Science, 1-1 Namiki, Tsukuba 305-0044, Japan. [9] Kavli Energy NanoSciences Institute at the University of California at Berkeley, Berkeley, CA 94720, USA. [10] Present address: Physics Department E20, Technical University of Munich, James-Franck-Straße 1, D-85748 Garching, Germany. These authors contributed equally: Jiong Lu, Hsin-Zon Tsai, Alpin N. Tatan. Correspondence and requests for materials should be addressed to J.L. (email: chmluj@nus.edu.sg) or to V.M.P. (email: vpereira@nus.edu.sg) or to M.F.C. (email: crommie@berkeley.edu)

Graphene's photon-like carrier dispersion provides fertile ground for testing exotic predictions of quantum electrodynamics, as well as for developing novel quantum electron optics[1]. Due to this relativistic behavior, electrostatic confinement of charge carriers in graphene is very different than that seen in more conventional materials[2,3]. Indeed, trapping electrons by placing point charges on graphene is formally analogous to trapping light by a gravitational field: something only possible near extremely dense matter[4]. Such localization, however, is possible for graphene around very strong Coulomb centers in the so-called supercritical regime[5–10], which allows a degree of localization otherwise impossible to achieve in pristine graphene. This behavior is formally equivalent to the supercritical collapse of atoms having ultra-heavy nuclei in quantum electrodynamics (QED)[11–15]. This atomic analogy, however, is only useful for charge distributions that can be approximated as a single-point charge. Here, we demonstrate a new supercritical regime, "frustrated supercriticality", that is accessible through careful arrangement of point charge distributions on a graphene surface. Frustrated supercriticality reflects an interplay between near-field and far-field electronic behavior for charge distributions that are globally supercritical but locally subcritical. Electronic behavior here is analogous to photons gravitationally trapped within a star cluster that has no black holes. The ability to charge and discharge such states via local electrodes raises the prospect of designing localized electronic states without compromising graphene crystallinity, and hence integrating them into extremely high-mobility nanoscale devices.

Demonstrating frustrated supercriticality in graphene requires the ability to position static charges with a level of precision currently unobtainable by conventional top-down lithography. We achieved the necessary precision via a bottom-up synthesis technique that yields charge-tunable, periodic, self-assembled one-dimensional (1D) arrays of $F_4TCNQ$ molecules on clean, back-gated graphene FET devices. STM spectroscopy (STS) measurements reveal that dilute charged arrays with large inter-molecule spacings $d \geq 10$ nm scatter surrounding Dirac fermions and induce no bound states in the nearby pristine graphene. For denser charged arrays with $d \leq 10$ nm, however, STS shows the emergence of a new quasi-bound state with an energy near the Dirac point. This state extends into the pristine graphene and is able to trap charge, as observed through spatially resolved charging maps. We are able to explain this behavior by modeling the combined array/graphene system via tight-binding calculations that take screening into account. Our simulations reveal that when intermolecular distance in a 1D array is greater than the graphene screening length then each molecule behaves like an isolated subcritical Coulomb center. For intermolecular separations less than the screening length, however, our simulations reveal the emergence of a new type of collective supercritical state with energy near the Dirac point. This *frustrated* supercritical state is seen theoretically even for systems composed of only two-point charges and the wavefunction spread scales with inter-charge separation. In the semiclassical limit, this behavior is shown to be nearly equivalent to a general relativistic treatment of trapped light.

## Results

### Structural characterization of $F_4TCNQ$ molecular arrays on graphene.

Our FET devices were fabricated by placing a CVD-grown graphene monolayer on top of a hexagonal boron nitride (h-BN) flake resting on an $SiO_2$ layer covering a doped Si wafer, the latter providing an electrostatic back-gate. $F_4TCNQ$ molecules (Fig. 1a) were used as the charge elements in this study because their charge state can be reliably switched on (negative) and off (neutral) via the back-gate, as demonstrated previously[16]. One-dimensional lattices of $F_4TCNQ$ were created using an edge-templated self-assembly protocol that allows highly precise alignment of individual molecules. The template consists of electronically inert 10,12-pentacosadiynoic acid (PCDA), a linear chain molecule that self-assembles into monolayer-high islands on graphene with perfectly straight edges[17] (Fig. 1a). As seen in the STM image of Fig. 1b, these islands display a regular moiré pattern with a period of $a = 1.92$ nm due to the lattice mismatch between graphene and the PCDA layer. When $F_4TCNQ$ is deposited at room temperature onto PCDA-decorated graphene/h-BN, we observe the preferential adsorption of individual $F_4TCNQ$ molecules at the PCDA island edge sites that correspond to a maximum in the moiré pattern (Fig. 1b, c). The precise moiré periodicity facilitates the assembly of 1D molecular arrays that remain strictly periodic over hundreds of nanometers, as shown in Fig. 1c. By controlling the dosage of $F_4TCNQ$ onto the surface, this edge-templating process results in tunable arrays that can exhibit periodicities ($d$) with unit cells having multiples of the moiré period $a$. $F_4TCNQ$ arrays with $d = 2a$, $3a$, $4a$, and $5a$ can be seen in Fig. 2a–d. Gate voltage control allows the molecules within an array to be toggled between negative and neutral charge states (Supplementary Fig. 1)[15]. The molecular charge state, for example, is negative when the gate voltage is 30 V for all molecular arrays down to (and including) a periodicity of $2a$.

### Probing electronic structure of graphene near charged molecular arrays.

We investigated how charged 1D molecular arrays

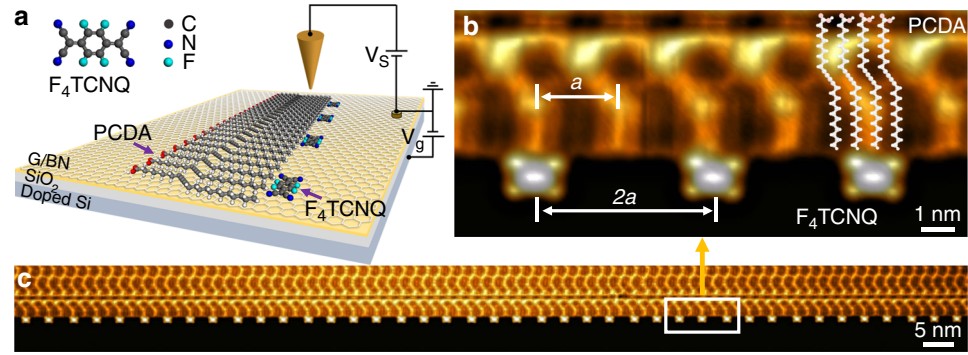

**Fig. 1** STM images of one-dimensional $F_4TCNQ$ molecular arrays. **a** Schematic illustration of edge-templated synthesis of $F_4TCNQ$ molecular arrays on a gated graphene FET device. **b** A close-up view of the PCDA edge-anchored $F_4TCNQ$ molecular array having a period of $2a$ ($a = 1.92$ nm is the moiré lattice constant of the PCDA monolayer on graphene). **c** STM image of an 80-nm long section of an atomically precise $F_4TCNQ$ molecular array having the $2a$ structure and anchored to the edge of a PCDA island. All STM images were acquired at T = 4.5 K

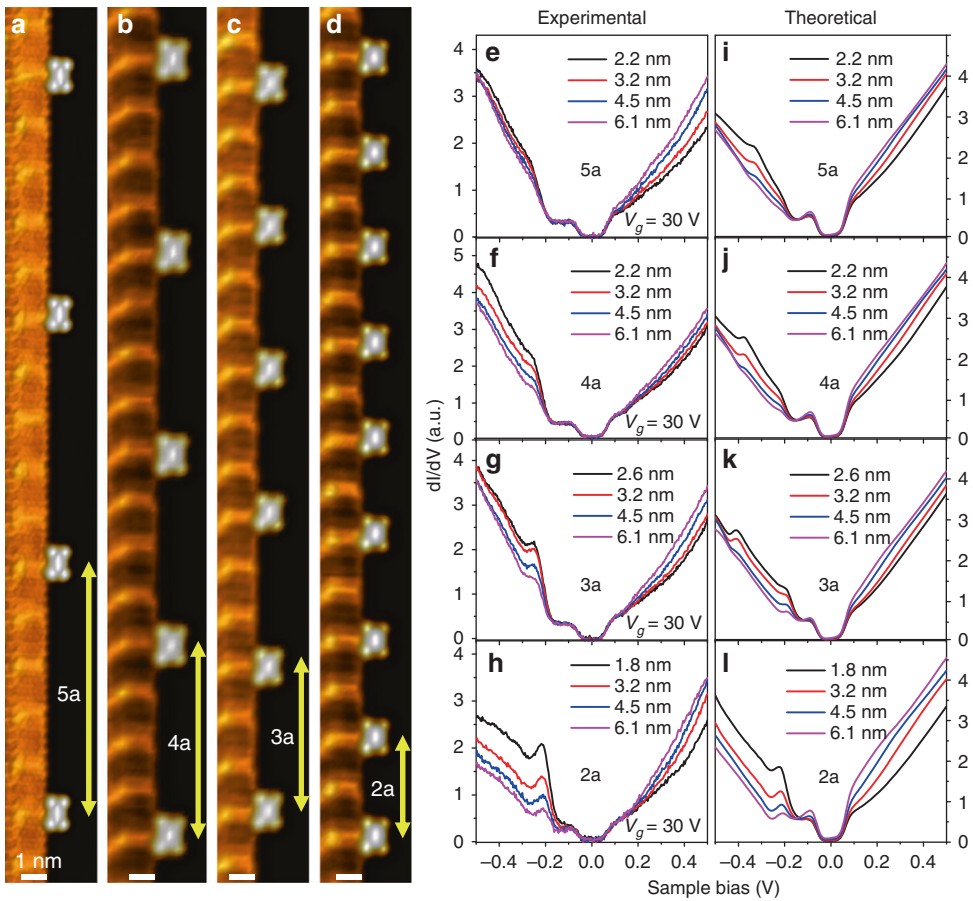

**Fig. 2** Emergence of supercritical features in 1D charged molecular arrays. **a–d** STM images of 1D $F_4TCNQ$ molecular arrays with tunable periodicity from *5a* to *2a* (the molecular arrays are anchored to PCDA islands at the surface of a graphene FET, and $a = 1.92$ nm is the PCDA/graphene moiré lattice constant). **e–h** $dI/dV$ spectra measured at different distances from the center of an $F_4TCNQ$ molecule along a line normal to the 1D array axis for charged arrays having different periods as shown in (**a–d**). All spectra were taken at the same back-gate voltage ($V_g = 30$ V) and tip height. **i–l** Theoretically simulated $dI/dV$ spectra for equivalent arrays of point charges on graphene at the same probing distances as in the experimental traces shown in (**e–h**). The calculation used an effective valence per molecule of $Z = 0.86$ and an effective Coulomb screening length $\lambda_S = 10$ nm, as described in the main text. All experimental data were obtained at T = 4.5 K

affect graphene's Dirac fermions by probing the energy-dependent local density of states (LDOS) in the vicinity of arrays having different periodicity. This was done by performing $dI/dV$ point spectroscopy on pristine graphene at different distances from the center of an $F_4TCNQ$ molecule along a line perpendicular to the charged array (Fig. 2e–h). All $dI/dV$ spectra exhibit a gap feature (~130 meV) pinned at $E_F$ (arising from phonon-assisted inelastic tunneling[18]) and another local minimum at $V_s \approx -0.18$ V for $V_g = 30$ V that indicates the Dirac point energy ($E_D$). $E_D$ is seen to lie 115 meV below the Fermi energy after accounting for the inelastic gap, corresponding to a carrier density of $n_e \approx 9.5 \times 10^{11}$ cm$^{-2}$ for $V_g = 30$ V. In arrays with a large intermolecular spacing of *5a*, the spectra at points adjacent to $F_4TCNQ$ molecules (Fig. 2e) exhibit the characteristic particle–hole asymmetry expected for an isolated *subcritical* negative charge (here, $Z < Z_C$, where $Ze$ is the charge on a molecule and $Z_Ce$ is the supercritical charge threshold; $Z_C = 1/2\alpha_0$ and $\alpha_0$ is the fine structure constant for graphene, see Supplementary Fig. 4)[5,6,16,19–22]. The graphene LDOS, however, changes substantially when the array period is decreased. As seen in Fig. 2f–h, the hole-side of the $dI/dV$ traces (i.e., $E < E_D$) develops a systematically higher spectral weight and a clear resonant structure near $E_D$ as the array period is reduced to *2a* (Fig. 2h). The resonance decays rapidly with distance from the array and fades beyond 10 nm (Supplementary Fig. 3). This new

feature cannot be attributed to a localized molecular orbital since $F_4TCNQ$ molecular states are more tightly bound and vanish at distances $s > 1.25$ nm from an $F_4TCNQ$ center (Supplementary Fig. 2), whereas the new resonance is observed over the range 1.8 nm $< s < 10$ nm.

Since isolated charged $F_4TCNQ$ molecules generate only a subcritical Coulomb potential[16], the development of a resonance near $E_D$ in more closely packed arrays suggests a collective effect whereby the array somehow surpasses the *supercritical* threshold and induces new quasi-bound states[7]. This hypothesis is supported by charging behavior observed near dense $d = 2a$ arrays, as seen in Fig. 3. Figure 3a shows a continuous region of the surface where the left side is imaged via an STM topograph (showing the 2a array) and the right side is imaged via a $dI/dV$ map that shows electronic structure in the pristine graphene to the right of the array for $V_S = -0.12$ V and $V_g = 20$ V. Sharp rings are seen on the right that are indicative of charging behavior (similar rings have been seen previously by STM due to the charging of adsorbed molecules and defects on various surfaces[23–27]). The rings of Fig. 3a, however, are centered away from the molecules on the pristine graphene, indicating that they arise from states localized in the pristine graphene rather than in the molecular orbitals.

This charging behavior can be better seen in the gate-dependent $dI/dV$ point spectra of Fig. 3b, acquired with the

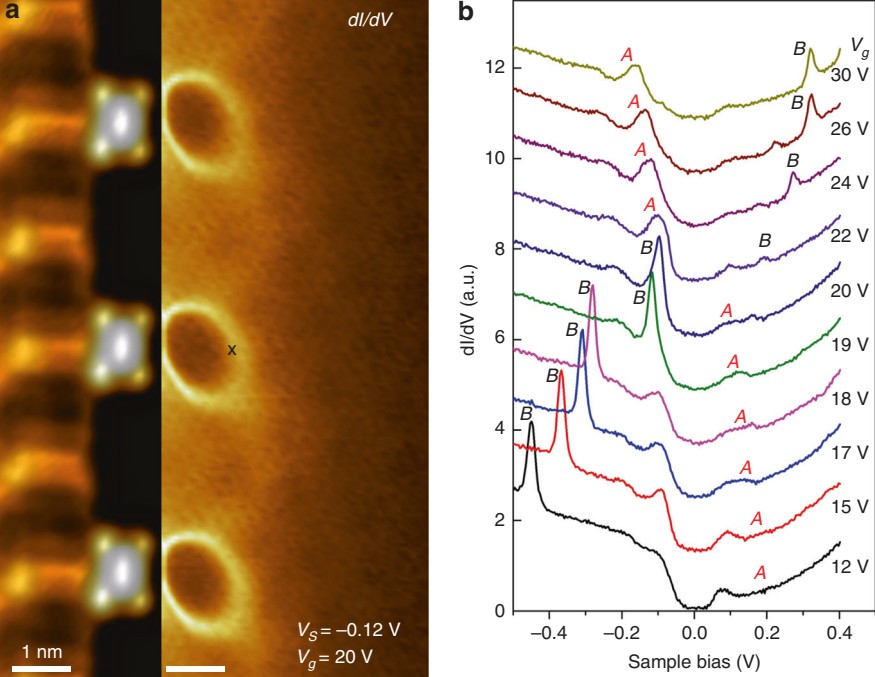

**Fig. 3** Gate-dependent charging behavior of supercritical quasi-bound state. **a** Left side: STM image of a portion of a charged 2a F$_4$TCNQ array. Right side: d$I$/d$V$ map of the pristine graphene region adjacent to the array shows charging rings in the near-field region ($V_g = 20$ V, $V_S = -0.12$ V). **b** Gate-dependent d$I$/d$V$ spectra acquired at the position marked "x" in panel (**a**). The supercritical resonance is labeled "$A$", and the corresponding tip-induced charging/discharging feature is labeled as "$B$"

STM tip held at the edge of the ring marked in Fig. 3a. The peak marked "$A$" shows the new graphene resonance induced by the charged molecular array as seen in Fig. 2h. As $V_g$ is lowered from $V_g = 30$ V to 24 V, this feature moves up in energy, as expected for a density-of-states feature when $E_F$ is lowered by reduction of $V_g$. An additional peak marked "$B$" can also be seen that moves opposite to $A$ as $V_g$ is lowered, indicating that it is a charging peak rather than a density-of-states feature[16,27]. For $V_g > 20$ V, peak $B$ is caused by the discharging of state $A$ (i.e., by loss of an electron) as it is pulled *above* $E_F$ by STM tip-induced local gating. For $V_g <$ 20 V, peak $B$ jumps across $E_F$ and continues to move down in energy, as expected for a charging peak since state $A$ has now crossed to the other side of $E_F$ (the empty state side) and must be pulled *below* $E_F$ to become charged (i.e., by gain of an electron). The charging behavior observed for this new graphene state confirms its localized nature (see Supplementary note 6 and Supplementary Figs 5 and 6 for additional details).

**Modeling the electronic structure of graphene near one-dimensional charge pattern**. To understand the microscopic origin of this new state, we set out to answer the question of how such a localized state might arise in pristine graphene from the effect of subcritical molecular Coulomb potentials. We started by calculating the LDOS for electronic states in the vicinity of a simulated 1D array of point charges on graphene. The simulation was performed by locating point charges at positions coinciding with the center of each molecule in the experiment and then calculating the LDOS via a recursive method[28] (see Supplementary Notes 7 to 11 and Supplementary Figs 7–17). Electrons in graphene were modeled using a single-orbital, nearest-neighbor tight-binding approximation[29,30] (Supplementary Note 7), and electrostatic screening was incorporated through the use of an appropriate dielectric function[15,31] (Supplementary Note 8). The resulting theoretically predicted d$I$/d$V$ traces are shown in Fig. 2i–l, beside the experimental ones of corresponding

geometry. To capture the inelastic phonon gap seen experimentally, we convolved the theoretical LDOS as described in ref. [32] (Supplementary Note 9).

Comparison of theory and experiment shows good agreement in all the key features: the overall particle–hole asymmetry, the marked increase of spectral weight for energies below $E_D$ as the array density is increased, the emergence of a clear resonance in the vicinity of $E_D$, and the rate of decay of the resonance with perpendicular distance from the array (see also Supplementary Note 10). Since our calculation included no perturbation to the graphene other than point charges, this confirms that the new structure in the d$I$/d$V$ curves is due to the collective Coulomb field of the charged F$_4$TCNQ array. Our best theory/experiment agreement is obtained for an effective valence per molecule of $Z = 0.86$ and a Coulomb screening length of $\lambda_S = 10$ nm (these values agree with previous estimates of $\lambda_S$ and $Z$ for isolated molecules adsorbed to graphene[16,33], see Supplementary Note 11). The estimated value of $\lambda_S$ is consistent with the experimental spatial extent of the resonant state, which is seen to disappear at distances $s > 10$ nm from an array (Supplementary Fig. 3).

In order to better understand the spatial distribution of this resonance state, we computed representative wave functions at energies within the resonance via exact diagonalization of the tight-binding model. As shown in Fig. 4a, a supercritical wavefunction is found that is confined to within a few nm of the array centerline and can thus be characterized as a quasi-localized state. This explains the strong, spatially decaying resonant state imprinted in the d$I$/d$V$ spectra of Fig. 2 as well as the fact that the resonance can be charged/discharged through local tip-gating (Fig. 3). The experimentally observed spatial offset of the charging circle to the side of the molecule (seen in Fig. 3a) can be explained by decreased tip-gating efficiency over the molecule's center due to the presence of highly concentrated negative charge on the F$_4$TCNQ molecules (see Supplementary Note 6 for details).

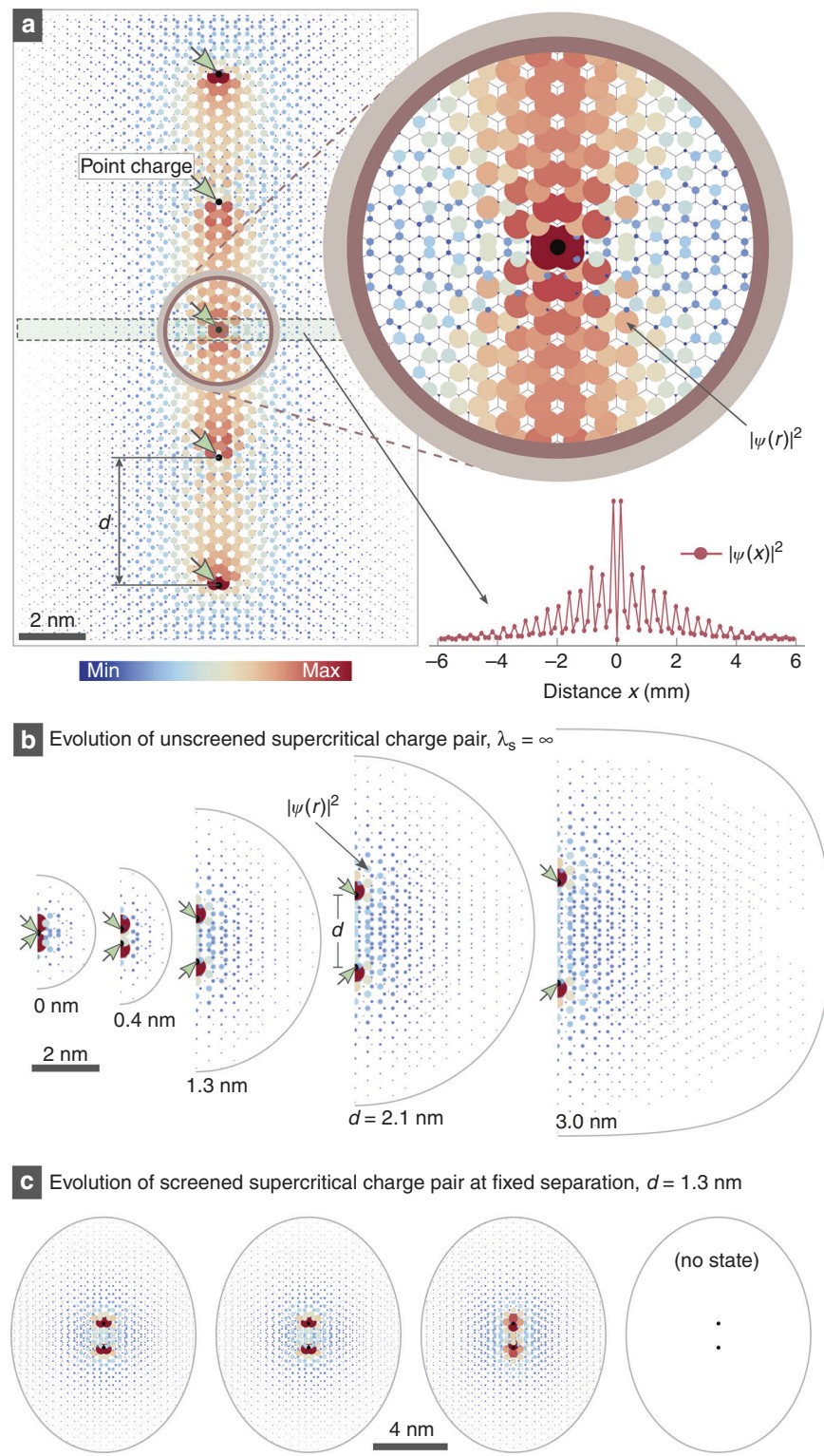

**Fig. 4** Theoretical wave functions for frustrated supercritical states. **a** Density plot of the wavefunction associated with a supercritical resonant state in graphene near the Dirac point obtained from exact diagonalization of the Hamiltonian discussed in the text (same parameters as in Fig. 2). Black dots mark the positions of the Coulomb centers used in the calculation and the colored disks reflect the state's local probability density, both through size and color. The charges are separated by $d = 3.8$ nm as in the experimental 2$a$ array and the total system has 16,000 carbon atoms spanning $19 \times 21$ nm$^2$ (the image shown is cropped). The top inset shows a close-up near the central charge, where rapid decay is visible against the underlying honeycomb lattice. The bottom inset shows the wavefunction cross-section along a line perpendicular to the array (boxed region, cf. Supplementary Note 9). **b** Wavefunction of the most bound supercritical state for a pair of *unscreened* charges at the following charge separations: $d = 0$ nm, $d = 0.4$ nm, $d = 1.3$ nm, $d = 2.1$ nm, and $d = 3.0$ nm ($Z = 0.8\,Z_C$). Each wavefunction is shown in the region where its value is at least 1% of its maximum. The characteristic wavefunction extension is ~$d$. **c** The same as (**b**) but with a fixed charge separation ($d = 1.3$ nm) and a varying screening length $\lambda_S$ as indicated. Supercritical states disappear for $\lambda_S \leq 3.6$ nm (cf. Supplementary Fig. 15d)

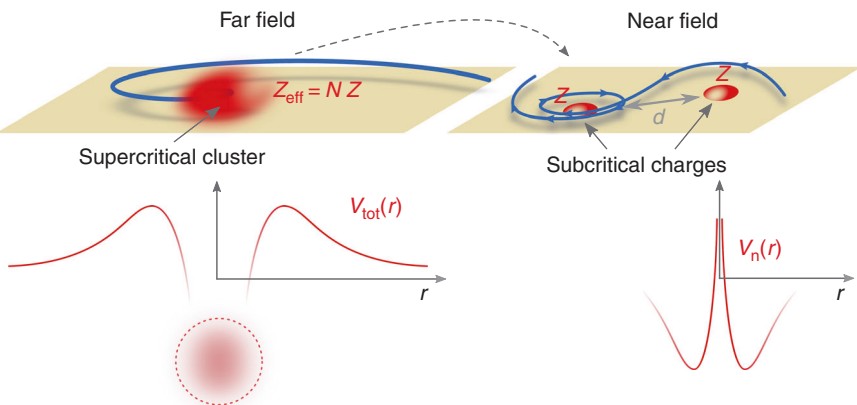

**Fig. 5** Far-field vs. near-field semiclassical trajectories for frustrated supercriticality: The far-field potential $V_{tot}(r)$ of a supercritical cluster (left) induces collapse because $N Z > Z_C$. Orbits here describe a collapsing spiral toward the charge cluster. In the near-field, on the other hand, each individual potential $V_n(r)$ is subcritical (right) and the orbits approach the charges without falling into them. The supercritical collapse is thus frustrated by the subcritical individual charges in the near-field. This is analogous to light rays gravitationally trapped by a dense cluster of stars (Supplementary Note 13.2)

The full 1D array simulations of Figs 2i–l and 4a reproduce our experimental data quite well, but they do not give us deep insight into the inner workings of frustrated supercriticality, including the interplay between near- and far-field behavior for Dirac quasiparticles interacting with distributed point charges. In order to gain a better intuition into this behavior, we analyzed quasi-localized states formed near globally *supercritical* charge distributions containing just two identical *subcritical* charges as a function of their separation[34,35] and screening length (each charge was given a valence $Z = 0.8 Z_C$). Figure 4b shows the results of exact diagonalization of the tight-binding model for this pair of charges with different separations, $d$, at the energy of the quasi-bound resonance (see also Supplementary Note 12). Localization of the quasi-bound state cannot be seen around any one charge center, because the near-field regions reflect the subcritical valence of the individual charges. Localization is seen rather in the far-field at distances $r > d$, where the aggregate charge of the interior can be seen as supercritical. As the two charges are pulled apart, the size of the quasi-bound state is seen to monotonically increase and push the far-field region outward from the origin. For unscreened systems, this process scales without limit as the subcritical charges are pushed apart to infinity.

The effect of screening on this process can be seen in Fig. 4c which shows the same two charges as in Fig. 4b, but for fixed separation $d = 1.3$ nm and different values of the screening length $\lambda_S$. The bulk of the wavefunction is seen to localize within $r \le 4$ nm, and so the state is essentially unchanged so long as $\lambda_S > 4$ nm. As $\lambda_S$ is reduced below 4 nm, however, the supercritical state rapidly quenches and the charge distribution reverts to subcriticality. The rapid quenching arises from two simultaneous effects. First, the two Coulomb potentials become physically separate as $\lambda_S$ approaches $d$ and, second, the supercritical wavefunction (which extends out a distance $d$) becomes constricted when the reduced screening length cuts into the potential that supports it. This explains why no signs of supercriticality are seen experimentally for our $d = 5a$ arrays, since the inter-charge separation in this case is on the order of $\lambda_S$. Supercriticality develops for denser arrays as the inter-charge spacing falls below the screening length ($d < \lambda_S$).

## Discussion

The contrasting behavior we observe here for the near- and far-field of a pair of subcritical charges each with $Z_c/2 < Z < Z_c$ can be summed up in a semi-classical description of graphene carriers under the effective potential, $V_{tot}(r)$, of a point charge distribution that is supercritical in the far-field but subcritical in the near-field. The supercritical regime is generally characterized semiclassically by the existence of a finite potential barrier that traps carriers on the charge distribution side of the barrier (details in Supplementary Note 13.1). For a carrier in the far-field, the potential appears supercritical, as schematically represented in Fig. 5 (left), and the relativistic nature of graphene renders the potential singularly attractive, namely $V_{tot} \sim -1/r^2$. The centrifugal barrier is unable to counterbalance this singularity and the orbits become collapsing spirals (see Supplementary Fig. 18)[19,36,37]. The far-field singularity, however, is removed at short distances from individual charge centers since $Z < Z_C$. The about-to-collapse far-field orbit is thus modified when it reaches the near-field of the cluster, where collapsing orbits can't exist due to the centrifugal barrier. The "collapse to the center" that seemed inevitable in the far-field is thus frustrated, as sketched in Fig. 5 (right), by the regular near-field behavior. Instead of collapsing, the particle becomes trapped within a region that extends out to $\sim d$, the distance between charges.

A useful analogy for this electronic behavior is the propagation of light near cosmic mass distributions accroding to general relativity. If a single, continuous mass distribution is compact enough that its spatial extent lies within the Schwarzschild radius, $R_{SC}$ (i.e., a black hole, see Supplementary Note 13.2), then light will be gravitationally trapped and inexorably fall through the event horizon toward the center[4], precisely the analogue of electronic supercritical collapse in the presence of a *single* supercritical impurity (i.e., graphene carriers here are mapped onto photons and the supercritical charge onto a black hole). On the other hand, if a mass distribution consists of isolated masses that each have no event horizon (e.g., a star cluster) but that extend close to $R_{SC}$ of the aggregate, then photons incident from outside of $R_{SC}$ can be trapped gravitationally in an extreme case of gravitational lensing. Unlike near a black hole, the photon's orbit will not end with a fall onto one of the stars, but will rather meander endlessly within the cluster, permanently bound by its gravitational field. This is completely analogous to the frustrated supercritical orbits of graphene charge carriers that remain trapped in the near-field of a cluster of subcritical charges whose total charge $> Z_C$ (cf. Supplementary Fig. 19 and Supplementary Note 13.3).

In conclusion, we have discovered a new physical regime of frustrated supercriticality that is accessible experimentally due to

advances in our ability to create atomically precise mesoscopic arrangements of Coulomb potentials on graphene. This creates new opportunities for manipulating charge states in high-mobility graphene devices and provides new insight into their behavior by analogy to astrophysical gravitational lensing of photons.

## Methods

**Graphene device fabrication.** A back-gated graphene/h-BN/SiO$_2$ device was prepared by overlaying CVD-grown graphene onto hexagonal boron nitride (h-BN) flakes exfoliated onto a SiO$_2$/Si substrate. h-BN flakes were exfoliated onto heavily doped silicon wafers and annealed at 500 °C for several hours in air prior to graphene transfer. The graphene was grown on copper foil by the CVD method and transferred to the h-BN/SiO$_2$ substrate via a poly methyl methacrylate stamp[38]. Electrical contact was made to the graphene by depositing Ti (10-nm thick)/Au (30-nm thick) electrodes using the stencil mask technique.

**STM/STS measurements.** STM/STS measurements were performed under UHV conditions at T = 5 K using a commercial Omicron LT STM with tungsten tips. STM topography was obtained in constant-current mode. STM tips were calibrated on a Au(111) surface by measuring the Au(111) Shockley surface state before all STS measurements. STS was performed under open feedback conditions by lock-in detection of an alternating tunnel current with a bias modulation of 6–16 mV (r.m.s.) at 400 Hz added to the tunneling bias. WSxM software was used to process all STM images[39].

**Theoretical modeling.** The theoretical calculations are described in detail in the following sections of the supplementary information: Tight-binding model of the charged arrays in graphene (Supplementary Note 7), Simulated d$I$/d$V$ curves from the bare LDOS calculations (Supplementary Note 8), Decay of the computed LDOS with distance (Supplementary Note 9), Screened Coulomb potential (Supplementary Note 10), Estimation of the effective potential parameters (Supplementary Note 11), Supercritical threshold of an array of subcritical charges (Supplementary Note 12), Effective radial potentials (Supplementary Note 13).

## Data availability

The data that support the findings of this study are available from the corresponding author on reasonable request.

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

## Acknowledgements

This research was supported by the Director, Office of Science, Office of Basic Energy Sciences, Materials Sciences and Engineering Division, of the US Department of Energy under contract no. DE-AC02-05CH11231 (Nanomachine program-KC1203) (STM imaging and spectroscopy), by the Molecular Foundry (graphene growth, growth characterization), and by the National Science Foundation grant DMR-1807233 (sample fabrication). J.L. acknowledges support from the Singapore Ministry of Education grant under R-143-000-A06-112 (data analysis). H.-Z.T. acknowledges fellowship support from the Shenzhen Peacock Plan (Grant no. 827-000113, KQJSCX20170727100802505, KQTD2016053112042971). V.M.P. acknowledges support from Singapore National Research Foundation under its Medium-Sized Centre Programme (theory formalism development), and by the Singapore National Research Foundation award "Novel 2D materials with tailored properties: beyond graphene" NRF-CRP6-2010-05 (charged array simulation).

## Author contributions

J.L., H.Z.T. and S.W. designed and performed the experiments and analyzed the data. A.N.T. and V.M.P. performed the theoretical modeling and analysis, with A.N.T. computing the simulated LDOS and dI/dV curves, and V.M.P. performing the exact diagonalization and analytical calculations. A.A.O., D.W. and A.R. helped with the experiments and gave technical support and conceptual advice. A.Z. and E.P. facilitated the sample fabrication. K.W. and T.T. gave technical support and grew h-BN for the device. M.F.C. supervised the experiments and data analysis. J.L., H.Z.T, V.M.P. and M.F.C. wrote the paper.

## Additional information

**Competing interests:** The authors declare no competing interests.

