## [Peer Review File · Nature Communications]

Reviewers' comments:

Reviewer #1 (Remarks to the Author):

The authors present an experimental and theoretical study of atomic collapse in the presence of an array of charges. Atomic collapse is a problem that dates back to the problem of relativistic atoms which is almost a century old problem and which received renewed interest since the study of graphene. The present work represents an important advance in this field of research. The bottom-up synthesis using F4TCNQ molecules to realize atomically positioned tunable charge arrays in graphene is very interesting and new. Atomic collapse state (i.e. supercritical confinement) is realized using two arrays of charges (modelled as two charges separated at a distance d) that are each subcritical is new and the physics of it is clearly presented. Thus the authors have realized a system with locally subcritical charges that are globally supercritical which is intuitively as expected.

Before the manuscript is published I would ask the authors to consider the following comments and suggestions:

- 1) The supercritical regime can also be realized without the use of supercritical charges. That was recently demonstrated by Y. Jiang et al [Nat. Nanotechnology 12, 1045 (2017)] using a sharp AFM tip which is able to induce a local deep potential. This reference can be added to put the problem in a broader perspective.
- 2) Using the edge-templated self-assembly approach 1D moire pattern is introduced. I am wondering if this does not lead to any strain into graphene which would result in an inhomogeneous pseudo-magnetic field that may influence the picture.
- 3) Only a single resonance, i.e. quasi-bound state, is observed in contrast to Ref. 8 where multiple resonances were observed. Do the authors expect to observe more resonances if the induced charge (and/or ' d ' is made smaller) is increased?
- 4) At several places 'trapping' and 'quasi-bound' states are used. It should be made clear that the charges are NOT trapped. One has only quasi-bound states because of the Klein tunneling effect.
- 5) It would be interesting to present a figure with a plot of the position of the different peaks in Fig. 3(b) as function of V_g where curves for the Fermi energy and the Dirac point are added. In the same figure the results from theory can be added. The experiment theory comparison presented in Fig. 2 is not very informative.
- 6) S.12.2.1 in the supplementary material: in this section the diving of energy states in the lower continuum is discussed. As suggested in this section more insight in supercriticality can be obtained by assuming a finite size system and tracking how the states evolve as the impurity strength changes. This method was already suggested by Pomeranchuk in 3D QED [See reference: I. Pomeranchuk and Y. Smorodinsky, Sov. Phys. Usp. 9, 97 (1945)]. It was also shown in recent publications [See: R. Van Pottelberge, et al, Phys. Rev. B 95, 245410 (2017)] [See: R. Van Pottelberge,

et al, Phys. Rev. B 97, 207403 (2018)] that in the case of a finite size system the resonances are replaced by a series of anti-crossings which are the signature of atomic collapse. From this perspective it is strange to associate the critical charge with the crossing of $E = 0$ since nothing special happens at this point. The interesting physics occurs when the first electron state anticrosses with the first hole state and consequently hybridizes with it. On top of that it is known that the sharpness of this anticrossing is directly related to the width of the resonance in the limit of infinite sample size. This was also discussed by Pomeranchuk in the case of 3D QED. The above behavior is in my opinion not really reflected in Figure S.13 of the supplementary material. It would be nice, if possible, instead of having a schematic figure to show a real finite size calculation showing the emergence of these anticrossings.

7) Next to Ref. [25] of the supplementary material, I think it is also appropriate to add the following reference since they performed similar calculations: Denis Klopfer, et al, Eur. Phys. J. B 87, 187 (2014). In this reference they showed, using the LCAO method, that using exact subcritical wave functions supercriticality can be achieved by sufficiently decreasing the distance between the two charges.

8) Regularized/unregularized Coulomb potential: in the theoretical calculations the array of charges is modeled by an array of point charges. As long as one is fairly deep in the subcritical region regularization is not required to obtain solutions and when the solutions are independent of any regularization parameter. As discussed in S11 of the supplementary material this is indeed the case in the performed experiment and consequently regularization is not required. However I am curious how the behavior changes theoretically if one approaches the sub/supercritical boundary by increasing the individual charges. Does the atomic collapse behavior become unfrustrated in some sense? It would be interesting if a small discussion regarding this question can

be added to the manuscript.

Reviewer #2 (Remarks to the Author):

This paper reports the observation of a new effect when one dimensional arrays of molecular charges are placed on graphene. This study follows previous work on the atomic collapse resonance corresponding to a large increase of the local density of states reported by some of the authors in reference 7. When these 1D arrays are densely packed, the tuning of the Fermi energy via electrostatic gating allows the system to be controlled from supercritical regime (i.e. when electrons are trapped around a strong Coulomb potential) to a frustrated supercritical regime (i.e. when screening length is smaller than the 1D array period) is found. The experiments are supported by tight binding calculations. The author claim that in these conditions electrons mimic light approaching a gravitational field and provide analogies with cosmology. The data is really nice, the article is well written and the results are clearly presented. The article is, to my opinion, of high

standard, well written do fit perfectly with the Nature Communications editorial guideline. Therefore I recommend it for publication in Nature Communications as it is.

Reviewers' comments:

Reviewer #1 (Remarks to the Author):

The authors present an experimental and theoretical study of atomic collapse in the presence of an array of charges. Atomic collapse is a problem that dates back to the problem of relativistic atoms which is almost a century old problem and which received renewed interest since the study of graphene. The present work represents an important advance in this field of research. The bottom-up synthesis using F4TCNQ molecules to realize atomically positioned tunable charge arrays in graphene is very interesting and new. Atomic collapse state (i.e. supercritical confinement) is realized using two arrays of charges (modelled as two charges separated at a distance d) that are each subcritical is new and the physics of it is clearly presented. Thus the authors have realized a system with locally subcritical charges that are globally supercritical which is intuitively as expected.

Before the manuscript is published I would ask the authors to consider the following comments and suggestions:

1) The supercritical regime can also be realized without the use of supercritical charges. That was recently demonstrated by Y. Jiang et al [Nat. Nanotechnology 12, 1045 (2017)] using a sharp AFM tip which is able to induce a local deep potential. This reference can be added to put the problem in a broader perspective.

In view of referee's comment, we have included this reference in the revised manuscript.

"Such localization, however, is possible for graphene around very strong Coulomb centers in the so-called supercritical regime⁵⁻⁹ [new reference] which allows a degree of localization otherwise impossible to achieve in pristine graphene."

2) Using the edge-templated self-assembly approach 1D moire pattern is introduced. I am wondering if this does not lead to any strain into graphene which would result in an inhomogeneous pseudo-magnetic field that may influence the picture.

That is indeed a pertinent comment. The following observations allow us to exclude such a strain-related picture.

First, we have mapped out the graphene lattice around the single molecule anchored on the edge of PCDA islands using non-contact AFM in the previously reported publication by Wickenburg, S. et al. [Nat. Commun. 7, 13553 (2016)]. That result shows that there is no significant strain observed for this adsorbate configuration.

Second, if the magnitude of a strain-induced pseudo-magnetic field is significant enough to affect the electronic properties, it is expected to have a characteristic signature in the dI/dV spectra [Science 329, 5991 (2010); Nature Comm. 3, 1068 (2012); Science 336, 1557 (2012); Nat. Comm. 3, 823 (2012)].

However, dI/dV data acquired close to the edge of PCDA without the decoration of any F4-TCNQ molecule is rather featureless.

3) Only a single resonance, i.e. quasi-bound state, is observed in contrast to Ref. 8 where multiple resonances were observed. Do the authors expect to observe more resonances if the induced charge (and/or 'd' is made smaller) is increased?

The following aspects must be considered in the context of this question, assuming that the individual charges remain subcritical.

a) At the theoretical level and ignoring screening of the Coulomb field contributed by each charge, one certainly has more than one supercritical resonance provided any subset of N charges separated by a distance d is supercritical ($NZ > 0.5$). However, whereas a single point charge of total valence NZ yields an infinite set of supercritical resonances that accumulate at zero energy and whose "Bohr radius" R_n extends systematically further from the charges with the approach to $E=0$, in a delocalized set of N individually subcritical charges, only those resonances whose typical "Bohr radius" lies in the far-field will survive. The main impact of this is that, if for a point charge of valence NZ one has resonances at energies $E_0, E_1, E_2, \dots, E_n, \dots, E_\infty=0$, in a delocalized cluster a number of those states ($n=0, 1, \dots, n_{\min}-1$) will be absent. The first, most bound one, appears at n_{\min} which is determined by the condition $R_n > Nd$. The primary effect of this is that the supercritical resonances will concentrate closer to the Dirac point ($E=0$), compared to having the same total charge on a single point. The first individual resonances are therefore seen much closer to each other in energy due to the exponential variation of E_n with n .

b) Still at the theoretical level, but now taking into account that the Coulomb potential from each charge is screened with a characteristic screening length λ , that imposes a second constraint so that, overall, one must have approximately $Nd < R_n < \lambda$. Since the energy of the supercritical resonances scales inversely with their localization length, these two bounds for the "Bohr radius" impose a corresponding energy interval within which supercritical resonances survive. This is seen explicitly in our numerical calculations. These considerations and additional details are discussed in supplementary section S12.1.

c) To describe the experimentally measured dI/dV , one must convolve the electronic local density of states (LDOS) as described in supplementary section S8, in order to simulate the effect of phonon-assisted indirect tunneling that leads to an apparent dI/dV gap of ~ 126 meV centered at zero bias voltage. However, in addition to this gap, this process introduces a natural broadening that will wash out fine structure of the originally calculated LDOS. This is illustrated in supplementary Fig. S9, where we compare the raw numerical LDOS with the simulated dI/dV after the convolution process.

Combined with the fact highlighted in (a) that the surviving supercritical resonances in a delocalized set of charges appear at energies closer to the Dirac point (and hence their inter-level separation is much smaller), in a practical case such as in our experiment, even if multiple resonances formally survive, one is not able to resolve them individually, even in the theoretical dI/dV , due to this broadening.

d) As discussed in the main text, our experimental analysis requires the samples to be slightly doped in order not to have the resonances superimposed with the inelastic gap in the dI/dV traces. The finite

carrier density leads to screening and, therefore, the screening of the Coulomb field contributed by each charged molecule cannot be avoided in our experimental conditions.

In summary, the conditions and constraints (both theoretical and experimental) for observing more than one resonance are well understood. In our experimental conditions, the main challenge is related to the compromise between the need to slightly displace the Dirac point from zero bias (i.e. to dope graphene) and screening of the Coulomb fields that should be minimized. At the moment, reducing the distance between charged molecules below the “2a” configuration is not possible in the current layout, because they become too close to each other and begin to interact electronically; the charged/discharged states are then not stable enough. Replacing F4-TCNQ with different molecules capable of holding more than a single unit of charge on demand is a promising alternative to investigate a stronger supercritical regime.

4) At several places ‘trapping’ and ‘quasi-bound’ states are used. It should be made clear that the charges are NOT trapped. One has only quasi-bound states because of the Klein tunneling effect.

The word “trapping” was only used when we discuss “*the trapping of electrons with electric field on graphene is analogous to the trapping of light by a gravitational field*”. In view of the referee’s comments, we have made sure to use “quasi-bound state” for the frustrated supercritical state throughout the main text.

5) It would be interesting to present a figure with a plot of the position of the different peaks in Fig. 3(b) as function of V_g where curves for the Fermi energy and the Dirac point are added. In the same figure the results from theory can be added. The experiment theory comparison presented in Fig. 2 is not very informative.

This is a good point. Unfortunately, the determination of the energy positions for all these features at all the gate voltages with respect to the Dirac point is not feasible due to the fact that (i) multiple components co-exist and (ii) the intensity of resonance peaks is significantly reduced above the Fermi level.

In figure 3b, we have labeled the peaks related to the resonance state and its associated charging peak as A and B respectively. We would like to point out that this figure aims to illustrate that the frustrated supercritical state can be charged or discharged by STM tip-induced local gating, analogous to impurity states in conventional semiconductors. Furthermore, the charging behavior observed for this new graphene state also confirms its localized nature, in good agreement with the distance-dependent behavior of resonance states as shown in Figure 2.

6) S.12.2.1 in the supplementary material: in this section the diving of energy states in the lower continuum is discussed. As suggested in this section more insight in supercriticality can be obtained by

assuming a finite size system and tracking how the states evolve as the impurity strength changes. This method was already suggested by Pomeranchuk in 3D QED [See reference: I. Pomeranchuk and Y. Smorodinsky, Sov. Phys. Usp. 9, 97 (1945)]. It was also shown in recent publications [See: R. Van Pottelberge, et al, Phys. Rev. B 95, 245410 (2017)] [See: R. Van Pottelberge, et al, Phys. Rev. B 97, 207403 (2018)] that in the case of a finite size system the resonances are replaced by a series of anti-crossings which are the signature of atomic collapse. From this perspective it is strange to associate the critical charge with the crossing of $E = 0$ since nothing special happens at this point. The interesting physics occurs when the first electron state anticrosses with the first hole state and consequently hybridizes with it. On top of that it is known that the sharpness of this anticrossing is directly related to the width of the resonance in the limit of infinite sample size. This was also discussed by Pomeranchuk in the case of 3D QED. The above behavior is in my opinion not really reflected in Figure S.13 of the supplementary material. It would be nice, if possible, instead of having a schematic figure to show a real finite size calculation showing the emergence of these anticrossings.

As the referee points out, the phenomenon of level diving into the opposite energy continuum (electron continuum for holes, and hole continuum for electrons) and its association with the supercritical instability is well known in the context of these problems. For that reason, in supplementary figure S15 of our original submission, we opted to present just the summary of the evolution of the energy levels as a function of different parameters (the individual charge Q , the number of charges N , and the Coulomb screening length λ_s). And, in the specific case of graphene, that effect in finite-sized systems has been explicitly scrutinized, both in gapless and gapped graphene, in the early references

V. M. Pereira, J. Nilsson, A. H. Castro Neto, Coulomb Impurity Problem in Graphene. Phys. Rev. Lett. 99, 166802 (2007).

V. M. Pereira, V. N. Kotov, A. H. Castro Neto, Supercritical Coulomb impurities in gapped graphene. Phys. Rev. B. 78, 85101 (2008).

(see Figs. 3c and 3 of each paper, respectively) which solve the Coulomb problem using the same tight-binding model on the honeycomb lattice that is used in our current manuscript (although, in these papers, the Coulomb interaction is not screened as is done in our current manuscript). Therefore, in our original submission, we opted to present only the sketch in Fig. S14 as a schematic reminder of the phenomenon in qualitative terms.

However, in light of the referee's remark, we agree that presenting the evolution of the levels explicitly for each of the representative cases we discuss is a useful complement to the results analyzed in Fig. S15. We have thus added the new supplementary Fig. S16, which shows the evolution of the levels closest to the Dirac point as a function of the same parameters varied in the corresponding panels of Fig. S15. In this way, we now present both the raw numerical data in Fig. S16, and the condensed summary in Fig. S15. (The schematic in Fig. S14 is maintained since it aids the introductory text in section S12.2.1 that discusses the effect in general terms.) Correspondingly, the text in supplementary section S12.2.1 has been updated to read

“Fig. S15 shows a number of representative examples of the application of this criterion to identify whether a multi-charge system is super or subcritical depending on the valence per charge Z , their separation d , as well as the impact of having a finite screening length λ_s in the Coulomb field. While the panels in Fig. S15 summarize each case in terms of the number of levels that have penetrated into the positive energy continuum, the respective panels in Fig. S16 show the detailed evolution of E_n for each case, in the vicinity of the Dirac point.”

Finally, the referee remarks that *“From this perspective it is strange to associate the critical charge with the crossing of $E = 0$ since nothing special happens at this point. The interesting physics occurs when the first electron state anticrosses with the first hole state and consequently hybridizes with it.”* This observation is absolutely correct as far as a finite-sized system is concerned. But we wish to point out that:

1. The purpose of our exact diagonalization calculations is to provide a definitive confirmation of both the existence of these quasi-bound states, their spatial distribution and decay profile, and that they are indeed associated with the “diving” process characteristic of supercritical physics. These calculations provide a theoretical perspective and qualitative insight that is independent of the LDOS calculations that we carried out on much larger systems (10^6 atoms). In this regard, we recall that our the LDOS is calculated, not by exact diagonalization of the tight-binding Hamiltonian, but by the numerically and memory-efficient recursive method described in supplementary section S7.3. Only thus are we able to simulate all the inter-molecule separations and STM probing distances investigated experimentally, with system sizes and dimensions exactly matching the experimental ones. Our aim is to theoretically interpret and describe the parameters specifically relevant for the supercritical regime observed in the experiments.
2. Precisely for the reasons indicated by the referee, estimating the supercritical threshold as the point where the first level-crossing occurs in a finite system overestimates the supercritical threshold in the thermodynamic limit: this is simply a finite-size effect due to the fact that a finite system has a strictly discrete spectrum, as seen and discussed in the two papers quoted above. The fact that graphene has a linearly vanishing density of states at zero energy makes this finite-size overestimation more severe because the levels closest to zero energy on either side of the energy axis are far apart by an energy that scales with the system size as $1/L$ (this is also explicitly seen in those two papers). This means that the threshold (for example, the critical valence charge in the single-impurity case) estimated by tracking the crossing of energy levels in a finite system depends strongly on the system size, and only asymptotically extrapolates to the supercritical threshold of the infinite system. On the other hand, even in a finite-sized system, levels can only cross zero energy beyond the supercritical threshold. As crossing $E=0$ occurs earlier than the first level-crossing, estimating the threshold by tracking the diving through $E=0$ provides a closer estimate of the critical value in the thermodynamic limit.

Therefore, even though indeed nothing special occurs when the first level crosses $E=0$ in a finite system, it provides a better criterion to estimate the supercritical threshold of the infinite system. Since in the manuscript and supplementary text we resort to such estimates to, among other things, validate the

Coulomb screening length used in the larger-scale LDOS simulations and observed experimentally, in the context of supplementary section S12.2 and supplementary Fig. S15 it is more appropriate to track the crossing of $E=0$.

Nevertheless, with the inclusion of our new supplementary Fig. S16 in the revised text, all these details of the spectrum in the finite-sized systems can be inspected explicitly, without ambiguity.

7) Next to Ref. [25] of the supplementary material, I think it is also appropriate to add the following reference since they performed similar calculations: Denis Klopfer, et al, Eur. Phys. J. B 87, 187 (2014). In this reference they showed, using the LCAO method, that using exact subcritical wave functions supercriticality can be achieved by sufficiently decreasing the distance between the two charges.

The new reference has been added to both the main and supplementary texts. In the main text, it now reads:

“In order to gain a better intuition into this behavior we analyzed quasi-localized states formed near globally supercritical charge distributions containing just two identical subcritical charges as a function of their separation [ref-1, ref-2] and screening length (each charge was given a valence $Z = 0.8 Z_C$).”

In the supplementary text, the statement indicated by the referee now reads:

“A similar conclusion has been obtained by studying approximate solutions of equation (39) in the particular case $N=2$ by either analyzing the asymptotic behavior of the solutions [ref-1], or solving the problem variationally in a LCAO (linear combination of atomic orbitals) scheme [ref-2].”

Where the two citations are

[ref-1] O. O. Sobol, E. V. Gorbar, and V. P. Gusynin, Phys. Rev. B 88, 205116 (2013).

[ref-2] D. Klöpfer, A. De Martino, D. U. Matrasulov, R. Egger, Eur. Phys. J. B. 87, 187 (2014).

8) Regularized/unregularized Coulomb potential: in the theoretical calculations the array of charges is modeled by an array of point charges. As long as one is fairly deep in the subcritical region regularization is not required to obtain solutions and when the solutions are independent of any regularization parameter. As discussed in S11 of the supplementary material this is indeed the case in the performed experiment and consequently regularization is not required. However I am curious how the behavior changes theoretically if one approaches the sub/supercritical boundary by increasing the individual charges. Does the atomic collapse behavior become unfrustrated in some sense? It would be interesting if a small discussion regarding this question can be added to the manuscript.

Throughout both the main and supplementary texts, we have almost exclusively concentrated in discussing the scenario where charge is individually subcritical. Not only is that the case of practical interest in these experiments but, generically, one expects it to be very difficult to have isolated point charges with a bare valence that exceeds the supercritical threshold. At any rate, one can argue that it is of interest to discuss the situation when each charge is driven across the individual supercritical threshold, even if only for completeness of the physical picture. We have thus added the following text to the new supplementary section “S12.3 – Beyond the individual supercritical threshold”.

“Although this case is not directly relevant for our current experiments, for completeness of the physical picture, imagine that one continuously increases the charge/valence on each molecule (Z), it is well understood [refs] that as soon as the supercritical threshold is overcome ($Z\alpha > 0.5$), each charge individually sustains an atomic-like series of resonant states whose respective wavefunctions are now localized within distances of the order of graphene's lattice constant. When such charges are arranged periodically, as in our experimental arrays, one will observe the formation of supercritical energy bands, in the same way that true atomic states lead to energy bands when they hybridize in a conventional crystal. In other words, supercritical bands emerge immediately after the individual supercritical threshold.

Experimentally, this would translate in distinctive behavior in comparison with our current regime where each charge is individually sub-critical: (i) there would appear supercritical states with a localization length smaller than the current one which is essentially dictated by the inter-charge separation; (ii) unlike what we see in Figs. 2, where the sharp resonance fades with increasing separation, sharp resonant levels should be detected independently of the inter-charge distance d (although their energy will vary with d); (iii) the fact that bands of supercritical states emerge beyond the threshold implies a relation between the characteristic width of the supercritical features in dI/dV and the width of the supercritical bands. Since that bandwidth decreases for larger inter-charge separations, the corresponding dI/dV features would become more sharp with increasing d . Hence, if one is capable of driving each charge individually across the supercritical threshold ($Z\alpha > 0.5$), these qualitative features in contrast with the subcritical behavior would altogether allow the experimental identification of that crossover.”

Reviewer #2 (Remarks to the Author):

This paper reports the observation of a new effect when one dimensional arrays of molecular charges are placed on graphene. This study follows previous work on the atomic collapse resonance corresponding to a large increase of the local density of states reported by some of the authors in reference 7. When these 1D arrays are densely packed, the tuning of the Fermi energy via electrostatic gating allows the system to be controlled from supercritical regime (i.e. when electrons are trapped around a strong Coulomb potential) to a frustrated supercritical regime (i.e. when screening length is smaller than the 1D array period) is found. The experiments are supported by tight binding calculations. The author claim that in these conditions electrons mimic light approaching a gravitational field and provide analogies with cosmology. The data is really nice, the article is well written and the results are clearly presented. The article is, to my opinion, of high standard, well written do fit perfectly with the Nature Communications editorial guideline. Therefore I recommend it for publication in Nature Communications as it is.

REVIEWERS' COMMENTS:

Reviewer #1 (Remarks to the Author):

The authors have convincingly replied to my different comments and have modified (and added) the manuscript accordingly.

I recommend the publication of the manuscript in Nat. Com.